# Research on Smart Tourism Oriented Sensor Network Construction and Information Service Mode

**DOI:** 10.3390/s222410008

**Published:** 2022-12-19

**Authors:** Ruomei Tang, Chenyue Huang, Xinyu Zhao, Yunbing Tang

**Affiliations:** 1School of Art and Design, Nanjing Forest University, Nanjing 210037, China; 2Research Center for Digital Innovation Design, Nanjing Forestry University, Nanjing 210037, China; 3School of Journalism, Fudan University, Shanghai 200433, China

**Keywords:** sensor data acquisition, smart tourism, data transmission, low energy consumption sensor, monitoring management system

## Abstract

Smart tourism is the latest achievement of tourism development at home and abroad. It is also an essential part of the smart city. Promoting the application of computer and sensor technology in smart tourism is conducive to improving the efficiency of public tourism services and guiding the innovation of the tourism public service mode. In this paper, we have proposed a new method of using data collected by sensor networks. We have developed and deployed sensors to collect data, which are transmitted to the modular cloud platform, and combined with cluster technology and an Uncertain Support Vector Classifier (A-USVC) location prediction method to assist in emergency events. Considering the attraction of tourists, the system also incorporated human trajectory analysis and intensity of interaction as consideration factors to validate the spatial dynamics of different interests and enhance the tourists’ experience. The system explored the innovative road of computer technology to boost the development of smart tourism, which helps to promote the high-quality development of tourism.

## 1. Introduction

In 2022, the World Federation of Tourism Cities (WTCF) and the Tourism Research Center of China’s Academy of Social Sciences jointly released the World Tourism Economic Trend Report (2022). The report showed that the global tourism revenue in 2021 was USD 3.3 trillion, equivalent to 3.8% of the worldwide GDP. Although this proportion has dropped significantly compared to before the Corona Virus Disease 2019 (COVID-19), it is still an essential part of the worldwide economy. In recent years, with the rapid development of Internet of Things technology, such as relying on big data, mobile devices, and sensors, more ways of data dissemination have been broadened. Physical media such as sensors have been embedded in daily life with unprecedented breadth and depth and extended to more fields. New industrial forms such as the smart city and smart tourism have emerged as the times require. Smart tourism is mainly composed of an intelligent information layer that collects numerical data, an intelligent exchange layer that supports interconnection, and an intelligent processing layer responsible for data analysis, visualization, integration, and smart use [1]. It aims to enhance the tourism experience through the most advanced information technology and big data [2]. In addition, promoting smart tourism development characterized by digitalization, networking and intelligence will help improve the service experience and promote the innovation of the smart tourism public service mode [3].

In this paper, we focus on the use of sensors and the construction of an intelligent system in Jiangsu Horticultural Expo. Sensors are installed in the scenic spots to collect information during the tour to realize the monitoring and management of the scenic spots. Jiangsu Horticultural Expo is rich in ecological environment resources, which is a successful demonstration of the “double-cultivation of cities.” It also has practical experience in applying digital-related technologies and has the technical foundation for turning digital research results into innovative tourism application products [1]. Therefore, it is necessary to develop a sensor platform that can monitor and analyze data in real-time, and use cloud computing technology to process and store the generated data to provide a scientific way for managers to solve problems [4].

## 2. Related Works

As the focus of scientific tourism management, smart tourism has attracted the attention of scholars. Governmental and academic institutions have in recent years attempted to design sustainable, technological, and efficient tourist cities to counter many of these problems. ( Gretzel et al.) [5]. At present, a lot of research has been conducted on smart tourism, involving the use of the latest technology. Aguilar et al. developed a new method to perform automatic billing functions in the cafeteria using neural networks [6]. Cacho et al. focused on developing intelligent travel guides for tourists, thus simplifying the travel planning process [7]. Kasnesis et al. have developed wireless acoustic sensors to identify and collect audio signals in cultural sites, and used them for data collection and the protection of cultural heritage [4]. Car et al. found that the adoption of IoT technology has improved the business processes and resource allocation of smart tourism [8]. Online social networks help to collect tourists’ preferences. French et al. have connected tourists in social networks, providing them with information, tour guides, and accommodation services [9].

In conclusion, the impact of the IoT and cloud computing on the tourism industry is disruptive. Communication technology has triggered significant changes in the field of the IoT, and the innovative iteration of sensors and cloud-based transmission has become essential features of smart tourism. Data-driven, computing-driven, and scene-driven formats have also become the inherent dynamism of smart tourism industry innovation [10]. At present, the research on improving the service quality of the tourism industry and promoting the transformation of intelligent tourism is still in the initial stage. Therefore, it is necessary to comprehensively improve the service function of the application computing system from the “end” application system and the “cloud” computing system of smart tourism.

Several studies on smart tourism information systems can be found in the literature. For example, by providing intelligent services to tourism departments such as transportation, hotels, and travel attractions and collecting feedback data, we can help build the integrity of the database system and formulate strategies to improve tourism management and services [11]. In addition, smart tourism information systems can be extended, and dynamic systems can offer efficient access to comprehensive sensor nodes and tourism platforms. This system provides a transaction settlement, intelligent guides, innovative marketing, and intelligence management of tourist attractions to support enterprises and tourism business [12,13]. In the dynamic environment of scenic areas, wireless sensor networks may receive a variety of environmental factors such as the electromagnetic field, temperature, humidity, noise, etc., and have a higher node failure rate and data loss rate than traditional networks. Significantly affected by energy consumption and size, the node configuration cannot achieve high sensor accuracy, and information loss is an inevitable problem. The first Reliable Data Aggregation Protocol (RDAT) secure data fusion algorithm based on a trust management mechanism was proposed by Ozdemir et al. [14,15]. By considering the energy efficiency factor, Liu et al. proposed an improved Reliable Trust-Based and Energy-Efficient Secure Data-Aggregation (iRTEDA) algorithm based on the RDAT algorithm [16]. In response to the above problems, an energy-efficient and reliable, secure data fusion algorithm, the Energy-Efficient protocol of Reliable Trust-Based Data Aggregation (ERTDA), is proposed in this paper, which can guarantee the reliability of the data transmission link and effectively extend the network life cycle. (Table 1)

In order to test whether the sensor nodes can realize the functions of wireless sensor network transmission and network convergence, this paper builds the sensor network platform shown in Figure 1. The environmental sensor node’s environmental monitoring function can collect information in three environments: temperature, vibration, and sound. The digital temperature sensor DS18B20, the three-week acceleration sensor ADXL345, and the digital MEMS sensor INMP441 are used, respectively. These sensors are small in size and easy for system integration. The output is digital, eliminating the need for noise reduction conversion circuits. Because the cooperation of multiple sensor nodes generally completes the monitoring of the target object, the sensor node sends data to the sink node. After receiving the data, the sink node tells the sensor node that it has received the data and ends the transmission. Because there is a time gap in the data transmission stage, the data is transmitted to the sink node by the multi-hop transmission strategy, which verifies the cooperation between different sensor types.

The rest of the paper is organized as follows. Section 2 introduces the research status of smart tourism and the construction of intelligent service systems in scenic spots. Section 3 describes the work method of the super-brain system and proposes the algorithm model of ERTDA and the process to cope with the scenic emergency pairs, the workflow of sensor signal reception, and transmission. Section 4 presents a few suggestions and application models for the challenge of low energy consumption of sensors. Section 5 presents results and discussion. This section introduced the main application results of the super-brain system and discussed the development of smart tourism.

## 3. Materials and Methods

The core of the intelligent sensor network platform for tourism is as follows:High performance data collection and fusion, comprehensive integration to promote tourism services, and information organization depth to monitor the spatial location, critical areas, and meteorological environment data of tourists in the environment in real time, and analyze and process the detection results.Enables intelligent sensors to operate with low energy consumption. At present, most wireless sensor nodes are powered by lithium batteries. Once they are used in environmental monitoring, battery replacement will become a big problem. However, the energy consumption of wireless sensor networks is generally consumed by nodes sending, receiving, and fusing data.

Therefore, another critical point in constructing intelligent travel sensor networks is to design low-power wireless transmission of environmental monitoring sensor nodes, which can effectively reduce the energy consumption in the wireless transmission process and significantly improve the running time of nodes.

Wireless sensor technology has triggered a new round of revolution in the field of the IoT. Each data acquisition point in the wireless sensor network is a small embedded system, which is the primary platform of the wireless sensor network, realizing intelligent data acquisition and cloud transmission. When nodes perceive the environment, they must convert non-electrical signals into electrical signals. A/D converters usually provide multiple analog channels, which can realize the conversion of numerous analog quantities. However, only one analog portion can be converted simultaneously, so the multi-way switch is used to select the analog amount in the structure shown in Figure 2. For example, in the clustering structure, the cluster head can fuse the data sent by the members in the cluster. At the same time, the common nodes can do preliminary digital filtering to improve the accuracy of data collection. The result of data processing can be temporarily stored in the node data storage module, sent to other nodes through the communication module, and transmitted to the sink node to realize remote data processing.

### 3.1. Data Collection

The ERTDA algorithm can effectively organize the captured nodes’ behavior of sending malicious messages in the network by calculating the trust value of the obtained target nodes compared with the threshold value, improving the security performance from within transmission network. The advantage of the ERTDA algorithm is that when the wireless sensor is used for environmental monitoring, it can focus on grasping and analyzing the monitoring data of a region rather than the data information of a specific node. It is an efficient, energy-saving, and reliable security data fusion algorithm. To evaluate the performance of the ERTDA algorithm, we run the iRTEDA model and the RDAT model in the same simulation environment and simulate and analyze the energy consumption rate and the life cycle of the network for the three models. In the parameter setting of the simulation, the whole wireless sensor network is set to contain 150 homogeneous sensor nodes randomly deployed in a 500 m × 500 m area.

We compared the number of dead nodes in adequate time for the three algorithmic models, and it is clear from Figure 3a that the number of dead nodes at the same time is much higher for the RDAT algorithm than for the iRTEDA and ERTDA algorithms. In the iRTEDA algorithm, the number of dead nodes is 50 when the network runs to 2500 s, after which there is a significant and rapid increase in the number of dead nodes. In the ERTDA algorithm, the energy factor of all nodes on the data link is taken into account, so the number of dead nodes does not reach 60 until the network runs until 3000 s, which effectively improves the life cycle of the transmission network.

Figure 3b shows the energy consumption ratio of the three models in the same simulation environment. As can be seen from the figure, the energy consumption ratio of the RDAT algorithm is much higher than that of the other two algorithms from the beginning of the network operation. When the network runs for 2500 s, the energy consumption of the RDAT algorithm network reaches 90%. In comparison, the iRTEDA algorithm and ERTDA algorithm only consume 43.5% and 38.6% of network energy, respectively, which greatly prolongs the life cycle of the network. The ERTDA algorithm considers the energy of other nodes in the data transmission link and the remaining energy of sensor nodes, which significantly slows down the rate of isolated nodes in the transmission network. When the network runs for 3000 s, the energy consumption ratio of the ERTDA algorithm only reaches 54.3%, which is lower than 72.4% of the iRTEDA algorithm.

Combining the simulation results and analysis of Figure 3a,b, we can accurately determine that the ERTDA algorithm model can effectively slow down the rate of dead nodes, reduce the proportion of energy consumption, and prolong the life cycle of the transmission network.

The ERTDA algorithm observes and monitors sensor nodes’ behavior using the Watchdog mechanism, including data collection, transmission, and fusion. For every other fixed period, the node records the received node data and calculates the trust value of the node by using the Beta distributed management model. Finally, all sensor nodes establish the trust value table through the Watchdog mechanism. The super-brain system needs to deal with four business systems: management, security, internet, and more than 30 kinds of real-time data. To ensure the accuracy of data collection, we introduce the ERTDA algorithm to make a correct or wrong binary evaluation judgment on the behavior of sensor nodes by using Beta distribution and define the Beta probability density function by gamma function Γ:(1)$$Brta\left(\varphi |\alpha ,\beta \right)\frac{\Gamma \left(\alpha +\beta \right)}{\Gamma \left(\alpha \right)\cdot \Gamma \left(\beta \right)}\varphi^{\alpha -1}{\left(1-\varphi \right)}^{\beta -1},$$
where 0 ≤ *φ* ≤ 1, *α* > 0, *β* > 0, here is the probability of the behavior or event represented by the sending parameter *φ*. However, when the constraint *α* < 1 is satisfied, *φ* ≠ 0, and when *β* < 1, *φ* ≠ 0, the expected value of the Beta probability distribution density function is:(2)$$Ε\left(\varphi \right)=\frac{\alpha }{\alpha ÷\beta }\;,$$

In the ERTDA algorithm, it is assumed that sensor node i and sensor node j observe each other’s state. The parameters *α* and *β* are represented by positive correct behavior m and negative wrong behavior n. Once the trust degree of the possibility of an event happening in the entity in the future is obtained, it will be:(3)α=m+1, β=n+1
where m ≥ 0 and n ≥ 0. m is the number of correct behaviors of the target node j observed by the monitoring node i, n is the number of wrong behaviors of the target node j followed by the monitoring node i. The parameter φ is the reputation value of the target node, therefore it represents the probability that the reputation value of the target node takes different values. It means the expectation of the reputation value of the target, that is, the er point. It indicates the trustworthiness of the node [17].

Under the guidance of wireless sensor network technology, more intelligent and miniaturized sensors bring a wide range of product information into the Internet of Things system. We adopt the improved ERTDA algorithm model to most likely assist sensor devices in efficiently processing user information. It strengthens the timeliness and accuracy of data processing, reduces data delay and broadband pressure, and realizes more accurate service and interaction with participants [10].

### 3.2. Perception Data Flow on the Intelligent Scene

The super-brain system focuses on the real-time monitoring of scenic area passenger flow and the timely issuance of passenger overload alert in terms of technical application. By analyzing the environmental capacity of scenic spots and setting the overload threshold of passenger flow, the data collected by sensors is converted into a tourist heat map to predict passenger flow. When the actual passenger flow in the scenic spot reaches the upper limit of passenger flow, it will provide a scientific basis for diversifying people in tourist-intensive areas. In addition, the temperature, humidity, wind speed, and other environmental factors in the scenic area are sorted out, and the meteorological changes in a short period of time are predicted to cope with potential meteorological disasters such as rainstorms, floods, and debris flow and ensure the safety of tourists’ lives and property.

In crowded scenes, pedestrian counting often cannot get high statistical accuracy because of unreliable detection. To solve this problem, in this paper, based on the use of Convolutional Neural Network (CNN) technology, we carry out pedestrian counting according to head detection. Firstly, we used the cascaded Adaboost detector to get the preliminary head proposals. Then, we used transfer learning technology to retrain CNN and, after that, the head classification model constructed by CNN and Support Vector Machine (SVM) was used to fine-recognize the head to improve the detection accuracy rate. Finally, the track association was used for tracking and counting the head targets. Experimental results show that our proposed method can locate a single pedestrian quickly and accurately, and the process has relatively high statistical accuracy [18].

Training sample: the positive sample used to train the CNN classifier model is consistent with the positive selection used to prepare the cascaded Adaboost detector;Training and detection process: After the test set images pass through the cascaded Adaboost detectors, many target areas are obtained. These target areas are input into the CNN model to get the final human head target;After the final head detection target is obtained by the CNN classifier model, the Euclidean distance is limited to the current detection target, the candidate-associated head matching area is obtained, and the associated head track is obtained. When the associated headway crosses the set detection line, a count is performed. During this process, signs of visitor movement are judged by the changes in sign position.

The CNN learning and training algorithm:

Input: training sample set:(4)$$S\left\{\left(x_{1},y_{1}\right),\left(x_{2},y_{2}\right),...,\left(x_{m},y_{m}\right)\right\}$$
while *m* is the total number of pieces:(5)$$y_{i}=\left\{-1,+1\right\}$$

Output: the final CNN classifier f(x).

### 3.3. Data Transfer and Events Processing

With the continuous development of 5G communication technology, the Internet of Things, with intelligent sensors as an essential bridge, has realized the intelligent sensing and cloud transmission of data information. Through the three-screen linkage of the command center, messages, events, and instructions can be efficiently issued and reported. For example, when a tourist is lost, different types of sensors will jointly search and provide helpful decision-making information in terms of a video surveillance query of the lost person, a prediction of its possible location, detection of nearby calling resources, etc., to speed up the handling of the incident of missing people (Figure 4).

Lost persons can call the emergency number of the park for user location, and the sensor can get the location information of tourists in time. Taking the maximum speed v of position movement in unit time as the radius of a circle as the in-situ sensor within the radiation range randomly samples n possible position points;The sensor observation service obtains real-time data of observation attributes or historical observations in a specified period. The trained classification prediction model predicts the next beacon node close to the sampling point per unit of time. It is summarized as the close beacon node. The unknown node is located in the intersection area between the maximum movement speed radius circle of the sampling point and the communication radius circle of the beacon node to which it belongs;The mobile sensor obtains the latest position data, or a position data sequence in a particular period of time, and uses the deflection direction of the mobile node to eliminate the impossible coordinate position points;Take the mean value of the remaining location points as the result of the location prediction of lost people. The video sensor links the data address and displays the observation data sequence in the form of graph and digital dynamic change.

There are n beacon nodes uniformly distributed in the park sensor network, each node has the same communication radius r and fails to cover the whole network, and each node has a signal-receiving device for receiving and measuring the signal strength from other nodes to that node. Let $P_{r}\left(d\right)$ denote the signal strength received by a receiver (j) at a distance d from source (i), $P_{r}$ is the transmitted power of the source, $G_{t}$, and $G_{r}$ denote the signal gain when sending and receiving, respectively, and (λ) is the electromagnetic wave wavelength. The relationship between them can be expressed by $F_{riis}$, the equation as:(6)$$P_{r}\left(d\right)={\left(\frac{\lambda }{4\pi d}\right)}^{2}P_{t}G_{t}G_{r}$$

Assuming that the magnitude of the signal strength measured by the signal source beacon node (i) itself is $P_{i}$, it will be:
(7)$$A\left[P_{i},P_{j}\right]=\frac{P_{r\left(d\right)}}{P_{i}}$$
where $A\left[P_{i},P_{j}\right]$ is the affiliation degree of an unknown node (j) to beacon node (i). The affiliation degree represents the ratio of the signal strength magnitude received from beacon node (i) at node (j) to the signal strength magnitude of beacon node (i) itself. From this, the node affiliation vector can be established as follows:(8)$$\underset{B}{\to }=\left\{ID,\;\left(x_{i},y_{i}\right),\;\;A\left[P_{i},P_{1}\right],A\left[P_{i},P_{2}\right],...,A\left[P_{i},P_{n-1}\right],A\left[P_{i},P_{n}\right],\;A\in \left(0,1\right)\right\}$$
where ID denotes the number of the ith beacon node and $\left(x_{i},y_{i}\right)$ is the coordinate position of beacon node (i). Since the lost visitor is in motion, each value A in its collected node affiliation vector $\underset{B}{\to }$ changes at all times, i.e., the closer $A\left[P_{i},P_{1}\right]$ is to 1, the closer the two nodes are. The unknown node’s motion unknown can be predicted according to the change law of A value, which improves the efficiency and accuracy of the calculation [19].

### 3.4. Intelligent Scenes and Interactive Experience

According to the scene theory, the scenes refer not only to spatial environmental scenes, but also includes the environmental atmosphere of behavior and psychology created by media information [20]. In the IoT environment, the connotation of the scene has been further extended. The collection, perception, processing, and analysis of scene elements, such as social relations, creates a sense of presence [21]. With the updated iteration of sensors, the amount of information collected by data increases exponentially. Personalized recommendations based on tourists’ preferences will become an essential channel for future travel consumption. The scene framework provides an ideal solution for personalized consumption experiences under artificial intelligence and big data technology [10].

The advantage of the super-brain system lies in integrating tourism with science and technology so that tourists can enjoy unmanned technology experiences and intelligent services in all aspects, such as ticket purchase, sightseeing, catering, shopping, transportation, accommodation, etc., and provide tourists with convenient and exciting consumption experiences. The IoT technology group has also improved the accuracy and efficiency of sensors, positioning systems, and big data transmission. For example, through all kinds of sensor devices and positioning systems, consumers’ activity information can be instantly collected, consumer services that meet the needs of popular culture can be customized, and personalized intelligent consumption scenes can be built. In addition to innovative services, the super-brain system is committed to improving the public service experience and providing brilliant hardware facilities and intelligent services. For example, in the visitor’s small program, you can make reservations and purchase tickets, check the arrival time of the sightseeing bus in real-time, and update the occupancy rate information of smart toilets within the play radius (Figure 5).

## 4. Discussion

Relying on the super-brain system, several sensor application products have been produced throughout smart tourism. Sensors have been widely used in scenic spot monitoring and management with different functions and scenes, such as panoramic spot running situation monitoring, tourist flow warning, tour route optimization, intelligent service experience, etc., and have achieved specific practical results. In short, integrating and combining sensors and tourist attractions is expected to create more comfortable tourism services and more excellent commercial value. However, it has been found that sensors still face particular challenges in their operation, maintenance, and cooperative processes. Meanwhile, sensors used in tourist attractions cannot make a unique and comprehensive standard according to different stakeholders and organizers, so it is necessary to promote and improve related research and practice continuously.

### 4.1. Range and Data Transmission Problems of Wireless Sensors

Wireless monitoring sensors solve the problem of collecting various environmental data, and they can realize long-term, timely, and reliable data collection and the wireless transmission of environmental information. However, from the existing practical experience, the power supply efficiency and system reliability of wireless monitoring sensors are limited. Wireless sensors also have obvious shortcomings, such as data fusion technology, power consumption, the dynamic topology of the network, limited node functions, fault tolerance, and so on [22]. The practical application in some scenic spots and the effect of analysis and prediction could be better. For example, the composition and structure of sensors in scenic spots are becoming more and more complex. In some areas, due to the extensive monitoring area and comprehensive sensor coverage, the monitoring devices are limited by wired or battery power supply, which makes it challenging to carry out large-area distributed installation, long-term use, and maintenance. At present, generating an energy model for sensor nodes can accurately predict the energy consumption of nodes, and it is an essential part of protocol development, wireless sensor network design, and Wireless Sensor Network (WSN) performance evaluation [23]. Currently, the practice of using wireless sensors to collect data is abundant, therefore three new schemes for reference are summarized for the research of the low-power sensing systems.

#### 4.1.1. Low-Power Device Selection

To reduce the system’s dependence on the power supply, the most direct and effective way is to reduce the power consumption of the system, that is, by designing a low-power energy management circuit. In low-power consumption design, selecting chip devices for hardware circuits is one of the most critical links [24].

The wireless sensor node is the smallest unit of the sensor network, composed of different structures according to other requirements. The technical bottlenecks of current wireless sensor networks are how to reduce the energy consumption of wireless sensor nodes and rationalize the energy of sensors. Due to the wide range of monitoring in the environment, it is necessary to deploy multiple nodes. Therefore, after ensuring that a single node can achieve a more efficient monitoring function, we should try our best to choose low-consumption and low-cost optimization and configuration. The wireless transceiver module exchanges the data sent by the processor module with other sensor nodes through wireless communication to meet information transmission requirements (Figure 6).

To ensure the long-term and stable operation of the monitoring point of the Expo site, it is necessary to select the low-power consumption sensor chip and the main control chip. Through comparison, it is found that using MSP430 as the main control chip can significantly reduce the system’s power consumption. At the same time, integrating the DS18B20 temperature sensor and the ADXL345 three-axis acceleration sensor can monitor the temperature and vibration information of the equipment in real-time and provide stable voltage output for the sensing system.

#### 4.1.2. Low-Power Circuit Design

Sensor nodes distributed in the Horticultural Expo are the basic units of the wireless sensor network. The design of low-power nodes can prolong the network’s life and effectively improve the operational performance of sensors. The sensor nodes contain the main power-consuming components of the circuit. These components only need to be on during operating hours, so in non-operating mode, the power supply to these components needs to be completely turned off to achieve low power consumption in the system [25].

Although the battery power supply of the sensor network is limited, the low-power circuit design has been used in empirical research to maintain the regular operation of the sensor and improve the working efficiency. Energy-aware routing (AODV routing protocol) will strive to keep most nodes running in their maximum lifetime. Each node with a high energy consumption rate and a short remaining life cycle should be shut down for a period of practice. The high energy consumption rate is determined by comparing the energy consumption rate of this node with other nodes. Closing a node will make the energy-aware routing protocol choose to replace the node or change the whole route to the destination node. This repeated process can distribute routing roles among most nodes, thus balancing the network’s energy consumption. The steps of the Ad hoc On-Demand Distance Vector Routing (AODV) energy awareness reason protocol are shown in Figure 7. These steps are as follows:

If a sensor node needs to transmit a message, it must check its routing table to find a way to the destination node. Thus, if a route can be found in the routing table, it forwards the message to the next node. Otherwise, the information is kept in the queue, and the source node sends routing request (RREQ) packets to its neighboring nodes to initiate the route discovery process;Before forwarding the message to the next hop, the energy consumption rate of the next hop is checked;If the energy consumption rate is high, the next hop will be closed for a specified period. The route will be removed from the routing table, which will result in starting the route discovery process again at the source node to find a new route to the destination node;

The synchronization mechanism of the wireless sensor can ensure that all nodes in the network wake up and sleep. On the one hand, it can avoid the waste of network nodes’ energy, and at the same time, it can ensure that all nodes in the system have the same energy consumption [26]. The flow chart of the time synchronization mechanism in this paper is shown in Figure 8.

#### 4.1.3. Dynamic Consumption Management Technology

Dynamic power consumption is when the load content is charged and discharged. Dynamic power consumption is the power consumption the digital circuit must calculate when it finishes its work, including flip power consumption and short-circuit power consumption. The short-circuit power consumption is the power consumption of the Complementary Metal Oxide Semiconductor (CMOS) when the Positive Channel Metal Oxide Semiconductor (PMOS) and the N-Metal Oxide Semiconductor (NMOS) transistors are turned on simultaneously. According to the principle of the capacitor charging and discharging point, the power consumption formula for inversion is shown as:(9)$$P_{switch}=a\cdot f\cdot c\cdot VVD2,$$
where *a* is the activity factor, *f* is the signal frequency, *c* is the load capacitance, and *VDD* is the supply voltage. The formula shows that switching power consumption is closely related to the load capacitance, activity factor, signal frequency, and supply voltage. Therefore, the supply voltage and frequency can reduce the flip power consumption. The input signal of CMOS is transformed by logic level, and the PMOS and NMOS are connected with the ground, resulting in a short-circuit current, whose consumption formula is shown as:(10)$$P_{internal}=t_{shart}\cdot f\cdot VVD\cdot l_{t},$$

The power consumption can be reduced by adjusting the duration of the simultaneous conduction of PMOS and NMOS. By synthesizing (9) and (10), the total dynamic consumption can be expressed as formula (11).
(11)$$P_{tatal}=a\cdot f\cdot c\cdot VVD^{2}+t_{shart}\cdot f\cdot VVD\cdot l_{t},$$

For environmental sensors, power consumption is not only a problem of energy consumption but is also affected by transmission and feasibility. Therefore, to reduce the sensor’s dynamic consumption during operation, a sensor dynamic power consumption management technology is proposed. According to the energy consumed in different working states, the duration of other active states is set to reduce power consumption. (Table 2) Dynamic consumption management technology can solve the workload situation in sensor nodes or the situation that idle nodes can not work again because of long-term existence.

It can be seen from the table that the conversion time of additional nodes in the intelligent sensor system is different in other active states.

In the dormant state, the consumption of sensor nodes is minimal. Still, the sensor nodes cannot perceive the surrounding environment, which leads to a lack of environmental monitoring information. At the same time, the transition from the dormant state to other modes takes a long time, which has a particular impact on the transmission of the data monitoring system. It should be designed and optimized in a dormant state, and the duration should be shortened as much as possible to make up for the deficiency of the transition;In the standby state, the central board node is in the dormant state, the wireless module is running, and the conversion time is relatively long, therefore it is necessary to sense and receive the information data of the node all the time. In the process of the transition from the receiving state to the standby state, the delay time is long;In the receiving state, the wireless sensor is in the state of receiving information data, and its consumption function is second only to the sending state;In the sending state, every part of the sensor system usually works, and all modules usually operate, therefore the consumption is also the largest. The running time should be shortened as far as possible based on ensuring the regular operation of the sensor system during optimization;

The number of tourists in the scenic spot will change with the tour’s content, so the wireless transmission route will significantly impact the power consumption in the network transmission process. Based on this, we optimized the routing protocol to reduce energy consumption in the network transmission process.

When the data transmission time in the stable transmission phase is longer than that in the cluster establishment phase, the energy consumption of network operation can be saved to a greater extent, and the data acquisition time can be increased. As shown in Figure 9, after measuring the geographical location and residual energy of cluster head nodes, the information collected by cluster head nodes with close distance and more residual energy is directly sent to the sink nodes. In the past, until all the data were transmitted to the sink node after this data transmission was completed, the system wiould conduct a new round of cluster establishment and stable transmission and re-plan the cluster head election and multi-hop transmission.

The working states of the above sensor nodes must be switched by combining software and hardware systems. The system power consumption can be reduced dynamically when the primary receiving, processing, and sending functions are realized.

### 4.2. Multi-sensor Collaboration Models and Data Fusion Issues

Many kinds of sensors have errors in data fusion, leading to a suboptimal estimation of events, resulting in low efficiency and significant processing problems. Just as the application of sensor data fusion to optimize the decision-making process in empirical research has produced remarkable practical results. According to the research literature, several ways are summarized to combine different data sources, such as decision-making, averaging, guidance, Bayesian statistics, and integration.

In terms of data fusion, data fusion by multiple sensors can accurately analyze and process data information while reducing the amount of data transmission for subsequent decision making and evaluation. In this paper, the data of multiple sensors are fused, which can reflect the mutual support of multiple sensors, thus avoiding the limitation of the measurement performance of a single sensor, thus improving the overall effectiveness and accuracy of the multi-sensor monitoring system. According to the test values of each sensor, the adaptive fusion algorithm finds the optimal weighting factor adaptively corresponding to each sensor. Under the condition of satisfying the minimum total mean square error, the fused result is optimized, thus meeting the needs of the environmental monitoring of the Expo, ensuring that the super-brain system of Horticultural Expo can quickly fuse data, coordinating and helping tourists in the scenic spot to meet various service needs (Figure 10) [27].

For any security system, the multi-sensor environment acquisition scheme is the key to overcoming the uncertainty caused by a single sensor. The cooperation mode and data fusion of different types of sensors help each group of sensors provide one kind of data for the system. The system fuses these heterogeneous data, thus realizing the mutual complement of sensor types (Figure 11).

The system used different types of data input as a part of the process to make improved decisions. The idea of this stage is to find the mutual complement of different types of input data. These provide an estimate (Kalman Filter) to predict the system’s state. Finally, this estimate is provided to the decision-making module, and the final system decision is obtained.

We compare the results with the commonly used data collection technologies, namely Envoy HTTP and Rabbit MQ. The results show that in the request or reply interaction (Figure 12a), when the number of traversed hops increases, the performance of CMOS will be better. The performance of CMOS forwarding realized by proxy is better at the local request rate (Figure 12b).

In the third test, following the same criteria as above, comparing and verifying that there are different complex infrastructures and network topologies between the two layers, there is routing capacity and delay in distributing increasing information among multiple users. The test results show that the sensor will be delayed even after passing through multiple intermediaries (Figure 13a), but the two solutions are relatively less affected by complexity. The results show that the proposed platform can process and memorize a large amount of data and re-compare, and provide extremely limited total delay (Figure 13b).

MATLAB 2020a software is used to simulate the experiment, and 2000 round experiments are carried out. The number of surviving nodes and energy consumption is compared to traditional sensors. (Table 3) Set 100 sensor nodes randomly distributed in a 10,000 square area, where the location of the sink node is in the center of the whole network (50m, 50m). The specific parameters are as follows.

### 4.3. Reference Achitecture of Personalized Service and Data Refinement of Sensors

According to the sensor’s specialized construction of different stakeholders and the organizer’s supervision system, it can reshape the tourism sensor of Horticultural Expo. Based on human-centered interaction and data collection, the personalized intensity experience of different tourists is generated. Sensors gather new service forms based on providing essential services for tourist attractions. They will combine them with vital services according to specific tourist profiles and current needs to create a common-view platform and customize personalized play experiences with different strengths for tourists. At the same time, based on the integration and enhancement of service capabilities, it will add additional value to producers and consumers. The data sensor is responsible for collecting, managing, and analyzing all the information provided by third-party suppliers, realizing the expansion and integration of data, and processing dynamic and non-dynamic real-time information in a timely and rapid manner [28]. For example, according to different tourists’ needs and other scene information in the venue, the dynamic route of cultural heritage, the active route of ecological restoration, the active route of sightseeing, the immersive interactive play route, etc., are customized. The application program combines tourists’ preferences with urban transportation networks and points of interest generated. Through the development of intelligent sensors in Horticultural Expo, this paper reconsiders and proposes smart tourism examples based on intelligent sensors: the extension and connection of clever technology to tourists’ gaze, interactive travel, thoughtful analysis and decision-making, immersion and authenticity, etc.

## 5. Conclusions and Future Work

In the research of this paper, it is found that data observation and resource fusion based on sensor technology can realize both big data collection for the whole area of Horticultural Expo and provide a reliable data source for the in-depth study of smart tourism. The data show that based on the trust management mechanism, the super-brain system proposed an efficient and energy-saving data fusion algorithm ERTDA, which effectively guaranteed the network security and extended the network running time. The hardware system and sensor technology are effectively combined to provide information technology support for treating emergency events in scenic spots and improve the overall management level of scenic spots. In addition, two referential solutions are summarized, aiming at the low-power consumption sensor system of the intelligent sensor in the Horticultural Expo, as well as the optimization method of multi-sensor and data fusion. In the deep integration of sensors and tourism, the performance of sensors should be optimized again to reduce the energy consumption of sensors.

As a recent example of “smart tourism”, the super-brain intelligent sensor system in Jiangsu Horticultural Expo is worth exploring for more possibilities. Although the intelligent system and operation mode of the “super-brain” system is analyzed in detail in this paper, other aspects of intelligent sensors are not discussed further. For example, personalized service customization, applying the super-brain system to the cultural value development of Horticultural Expo, popularizing and installing intelligent sensor systems in different tourism environments, and so on. In future research, consumer demand based on tourists’ preferences and personalized service will become an essential channel for smart tourism to upgrade its cultural industry format. Based on collecting and analyzing tourists’ data, the sensor group realizes the in-depth mining of cultural tourism consumption behavior. Drawing accurate portraits of tourists and perceiving tourists’ consumption preferences builds a scene setting that matches the tourists. Finally, it makes personalized recommendations on their consumption content. With the support of sensor technology, we realize the collection, perception, and processing of scenic scene elements, create multiple scene categories, develop and apply smart technologies to different tourism environments.

## Figures and Tables

**Figure 1 sensors-22-10008-f001:**
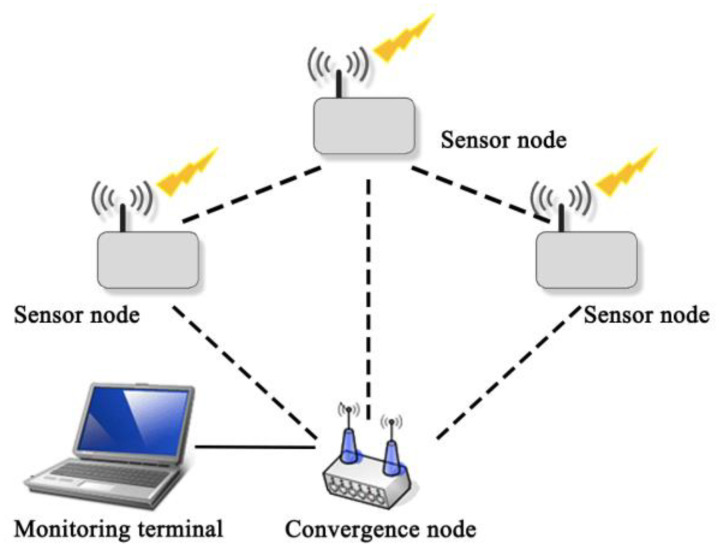
Structure diagram of sensor test platform.

**Figure 2 sensors-22-10008-f002:**
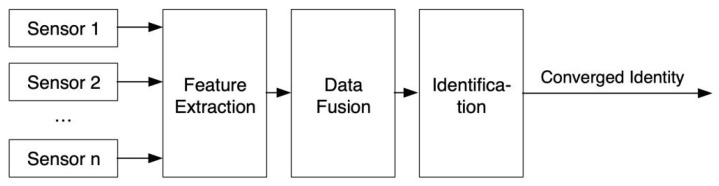
Data-level fusion structure.

**Figure 3 sensors-22-10008-f003:**
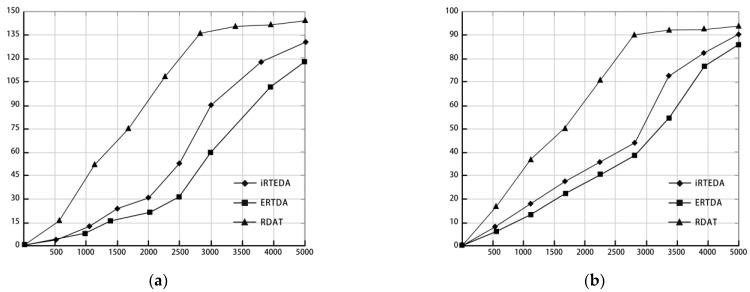
The consumption comparison of three algorithms. (**a**) The comparison of node death rates of three algorithm models. (**b**) The comparison of energy consumption rates of three algorithm models.

**Figure 4 sensors-22-10008-f004:**
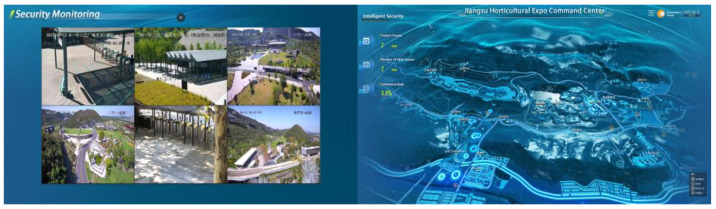
Super-brain system intelligent security system.

**Figure 5 sensors-22-10008-f005:**
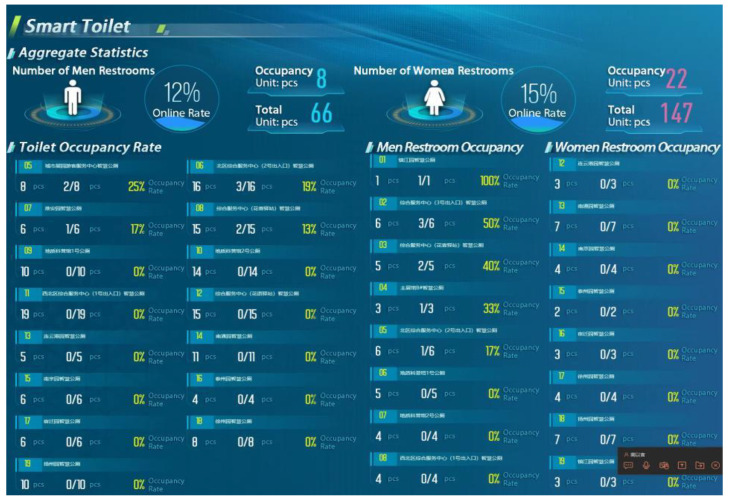
Smart toilet.

**Figure 6 sensors-22-10008-f006:**
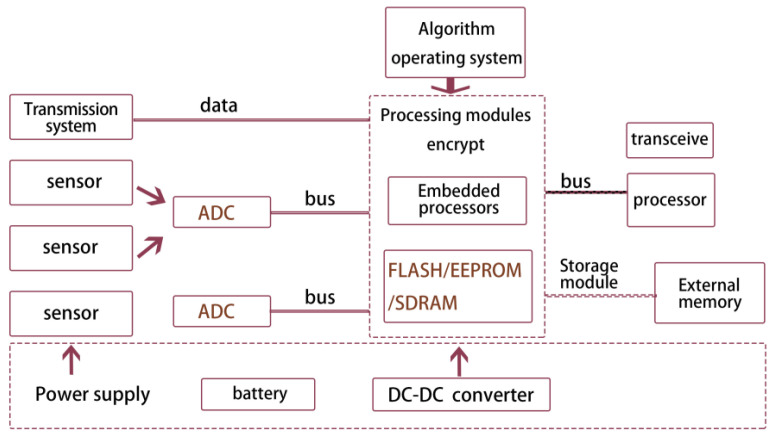
Hardware structure diagram of the wireless sensor node.

**Figure 7 sensors-22-10008-f007:**
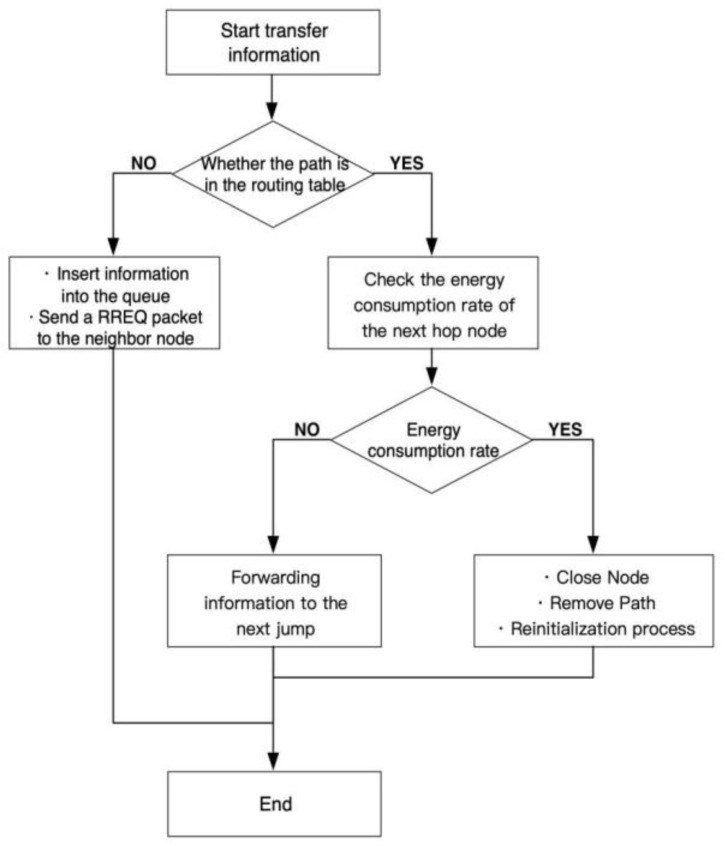
Steps of the AODV energy-aware routing protocol.

**Figure 8 sensors-22-10008-f008:**
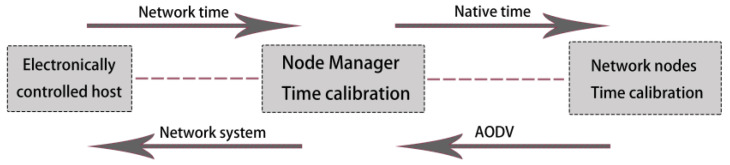
Time synchronization mechanism.

**Figure 9 sensors-22-10008-f009:**
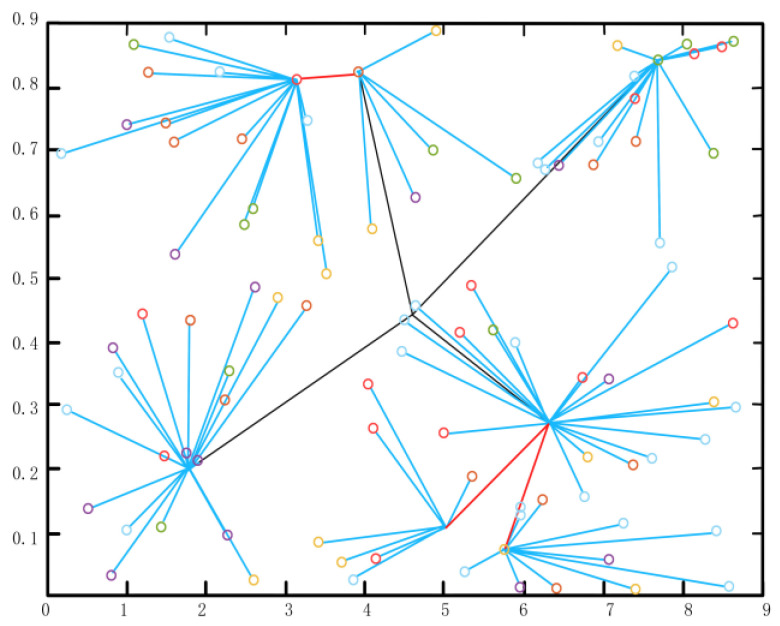
Network node clustering graph.

**Figure 10 sensors-22-10008-f010:**
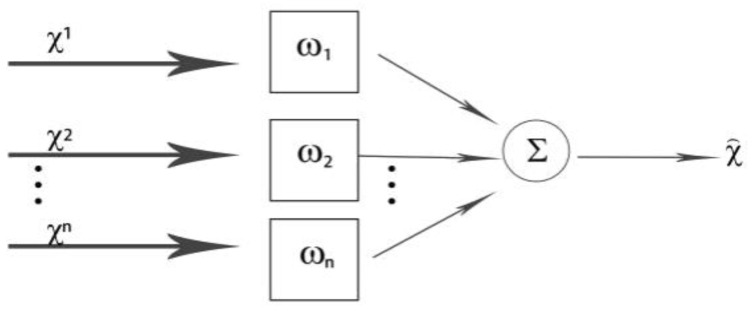
Adaptive weighted fusion algorithm model.

**Figure 11 sensors-22-10008-f011:**
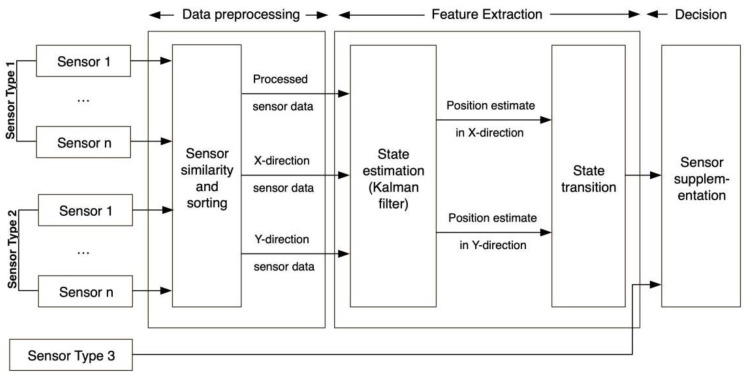
Complementary system block diagram.

**Figure 12 sensors-22-10008-f012:**
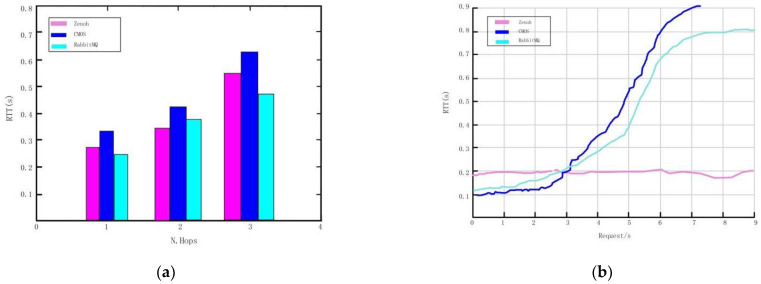
Data collection. (**a**) Graph of round-trip times for requests across network hops. (**b**) Request round trip time graph.

**Figure 13 sensors-22-10008-f013:**
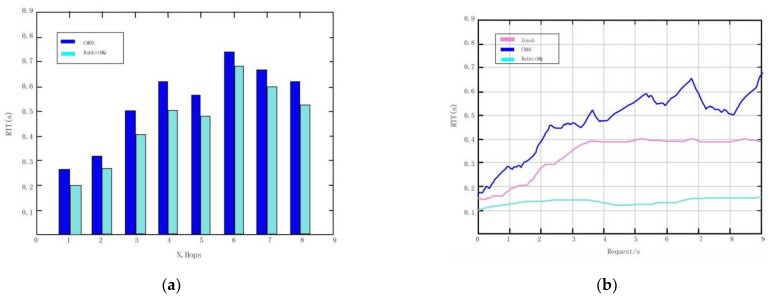
Data collection and comparison. (**a**) Sensor registration delay. (**b**) Data integration.

**Table 1 sensors-22-10008-t001:** Description of smart tourism application.

Methods	Application Domain	Sensing Accuracy	Energy Efficiency
Data Integration [11]	Management Services	-	-
IoT [13]	Marketing Services	-	-
RDAT Algorithm [14]	Data Transmission	Low	Low
iRTEDA Algorithm [16]	Data Transmission	Low	Low

**Table 2 sensors-22-10008-t002:** Sensor working status module table.

Conversion Duration	Control Module	Wireless Module	Functional Mode
10 ms	Rest	Standby	Resting State
20 ms	Rest	Open	Resting State
15 ms	Free	Open	Receiving State
15 ms	Run	Open	Sending State

**Table 3 sensors-22-10008-t003:** Simulation parameter table.

Parameter	Value
Convergent Node Position	(50 m, 50 m)
Number of Nodes	100
Network Area	100 m × 100 m
Packet Length	4000
Node Initial Energy $E_{0}$	0.5
Power consumption of free channel model signal amplifier $E_{FS}$	10
Consumption of Signal Amplifier in Multipath Fading Channel Model $E_{mp}$	0.0013

## Data Availability

Not applicable.

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
