# Peer review of "Research on Smart Tourism Oriented Sensor Network Construction and Information Service Mode"

_sensors, 2022, doi:10.3390/s222410008_

Round 1

Reviewer 1 Report

1.     Mention how much data is collected, stored and at what sample rate.

2.     As mentioned in the paper the new method is proposed of using sensor network observation data for building. So mention which sensors are used and how much data is collected in the abstract

and how your method is best then other previous method in the same application

3.     14 smart tourism information integration

4.     Check the format for the references and also add the latest papers and add more references.

5.     Add the Future work in the paper.

6.     The manuscript is not in the right format so kindly use the proper format. Read the guidelines carefully.

7.     In the Literature review, give the review in the table format with the drawback of existing methods and which method they have used.

8.      Many grammatical errors in the abstract and the full paper so kindly use the Grammarly help.

9.     Use equation editor to write the equations and also the parameters or the variables not defined in the text.

10.  If the figure is copied from somewhere, cite the paper in the figure caption.

11.  Font also changed in the entire paper. so kindly read the guidelines

12.  The abbreviations which have been used is not provided the full form of those short forms.

13.   Lot of spelling mistakes in the paper.

14.  Proper proofreading is necessary for the entire paper.

Reviewer 2 Report

Title: Research on Smart Tourism Oriented Sensor Network Construction and Information Service Mode

Manuscript IDsensors-2071116

Recommendation: Minor revision.

In this paper, the author proposes a new method of using sensor network observation data for building smart tourism information integration. The author developed and deployed earth observation sensors that collect data. These are transferred to a modular cloud platform that combines clustering technology and A-USVC location prediction methods to assist in emergency events. Interestingly, considering the attractiveness of the attraction to tourists, the system also incorporates human trajectory analysis and the strength of interaction relationships as inputs to validate the spatial dynamics of different interests and enhance the visitor experience. I recommend its publication with minor revision.

1. What is the most important thing in the construction of sensor network for intelligent tourism?

2. What are the advantages of super brain systems?

3. What are the disadvantages of the sensor? Is it over-dependent on power, energy, etc?

4. What are the future development ideas and directions of intelligent tourism management based on wireless perception? What are the challenges?

Reviewer 3 Report

-        The paper introduces the idea of implementing an ERDTA-based super brain system in scenic areas that collects data through low-energy sensors. This is an interesting idea that could be implemented in tourist-focused areas. But, there are areas within the paper that needs further clarification or improvements.

-        Firstly, the authors proposed the implementation of an ERDTA algorithm within the service system. Why the algorithm is proposed instead of other existing algorithms? The proposed usage of a collection of sensors that could assist tourists in an emergency or other activities seems to be supported by other algorithms. It is recommended that the authors provide further support on why the ERDTA algorithm is being proposed in this tourist situation.   
Besides that, the type of sensors used in collecting the data to support the proposed super-brain system. Further clarification on how the different sensors would support each other in ensuring that the super brain system could assist tourists.  How would the system determine that there are tourists within an area while ensuring that power consumption by the sensors are kept a low level?.

-        The purpose of the proposed system is unclear. The authors state that the system is to identify environmental capacity at scenic areas but it is unclear as to what purpose. Is it to manage the number of tourists within an area? Is it to ensure that during an emergency situation, the tourists’ safety is ensured while being evacuated? The authors need to be clear on this within the paper.  

-        Without a clear explanation on this usage, the discussion of the paper lacks the proper rigor required for the journal. Overall, it is a good paper but it could be improved with further information. 
